# Coverage and Determinants of Full Immunization: Vaccination Coverage among Senegalese Children

**DOI:** 10.3390/medicina55080480

**Published:** 2019-08-14

**Authors:** Abdur Razzaque Sarker, Raisul Akram, Nausad Ali, Zahedul Islam Chowdhury, Marufa Sultana

**Affiliations:** 1Bangladesh Institute of Development Studies (BIDS), Dhaka 1207, Bangladesh; 2International Centre for Diarrhoeal Disease Research, Bangladesh (ICDDR, B), Dhaka 1212, Bangladesh; 3School of Health and Social Development, Deakin University, Melbourne, Burwood, VIC 3125, Australia

**Keywords:** children, immunization, coverage, child health, Senegal

## Abstract

*Background and Objectives:* In line with the global success of immunization, Senegal achieved impressive progress in childhood immunization program. However, immunization coverage is often below the national and international targets and even not equally distributed across the country. The objective of this study is to estimate the full immunization coverage across the geographic regions and identify the potential factors of full immunization coverage among the Senegalese children. *Materials and Methods*: Nationally representative dataset extracted from the latest Continuous Senegal Demographic and Health Survey 2017 was used for this analysis. Descriptive statistics such as the frequency with percentage and multivariable logistic regression models were constructed and results were presented in terms of adjusted odds ratio (AOR) with a 95% confidence interval (CI). *Results:* Overall, 70.96% of Senegalese children aged between 12 to 36 months were fully immunized and the coverage was higher in urban areas (76.51%), west ecological zone (80.0%), and among serer ethnic groups (77.24%). Full immunization coverage rate was almost the same between male and female children, and slightly higher among the children who were born at any health care facility (74.01%). Children who lived in the western zone of Senegal were 1.66 times (CI: 1.25–2.21; *p* = 0.001) and the children of Serer ethnic groups were 1.43 times (CI: 1.09–1.88; *p* = 0.011) more likely to be fully immunized than the children living in the southern zone and from the Poular ethnic group. In addition, children who were born at health facilities were more likely to be fully immunized than those who were born at home (AOR = 1.47; CI: 1.20–1.80; *p* < 0.001), and mothers with recommended antenatal care (ANC) (4 and more) visits during pregnancy were more likely to have their children fully immunized than those mother with no ANC visits (AOR: 2.06 CI: 1.19–3.57; *p* = 0.010). *Conclusions:* Immunization coverage was found suboptimal by type of vaccines and across ethnic groups and regions of Senegal. Immunization program should be designed targeting low performing areas and emphasize on promoting equal access to education, decision-making, encouraging institutional deliveries, and scaling up the use of antenatal and postnatal care which may significantly improve the rate full immunization coverage in Senegal.

## 1. Introduction

Vaccines have the biggest success stories in the field of public health. Childhood immunization against vaccine-preventable diseases (VPD) is recognized as one of the most cost-effective programs to diminish childhood mortalities and morbidities across the world [1,2]. Every year, vaccination against VPDs prevents childhood illness, disabilities, and saves millions of children’s lives around the world [3]. It was estimated that, between 2010 to 2015 at least 10 million deaths were prevented globally and millions of more lives were protected from sufferings and disabilities associated with many infectious diseases [4]. Over the decades, remarkable improvements have been made toward the development of national immunization programs. Expanded Program on Immunization (EPI) of World Health Organization (WHO) has the major contribution to this global success [5]. EPI is a routine activity within the public healthcare system including mass immunization campaigns and door-to-door activities across the country aiming to increase the routine vaccine uptake. EPI was formally established in 1974 targeting childhood immunization against six vaccine-preventable diseases (i.e., diphtheria, pertussis, tetanus, poliomyelitis, measles, and tuberculosis) by 1990 [6]. Implementation of EPI has increased the vaccine coverage globally; in 2017 EPI achieved around 85% DPT (3 doses of diphtheria-tetanus-pertussis) coverage worldwide with 123 countries of at least 90% coverage. The global coverage of measles and polio was also high as near to 85% for both by the end of 2017 [7].

Along with the global success, Senegal also achieved significant improvements in the child immunization coverage and reduction of child morbidity and mortality. Senegal launched its EPI in 1979 with the support of international donors targeting immunizing against seven childhood diseases which was later increased to 11 in 2005 [8]. EPI program in Senegal recommends that a child should receive BCG (Bacillus of Calmette–Guerin) vaccine against tuberculosis soon after birth, oral polio vaccine (OPV), and the monovalent pneumococcal conjugate vaccine (PCV) at birth and at 6th, 10th and 14th week, and at 9th month. Since 2005, DPT vaccine had been replaced by pentavalent vaccine which contains a combination of five vaccines in one dose i.e., diphtheria, tetanus toxoid, pertussis, hepatitis B, and Hemophilus influenza type b vaccine (Hib) and recommended to be administered at the same time as the OPV or PCV at 6th, 10th, and at 14th week from birth (Table 1) [9]. Since measles is highly endemic and frequently affects infants in Senegal, routine measles vaccination is recommended at the 9th month of age with a second dose at 15th month. Additionally, a dose of yellow fever is also given at the age of nine months [9,10]. Regardless of successes, vaccine-preventable diseases are the leading cause of childhood mortality in Senegal while the under-five child mortality rate is still high as 45 per 1000 live births [11,12]. Further, the country experienced a rise in measles incidence in 2009 with high case fatality rate which indicates the necessity of increasing efforts in achieving full immunization coverage among children [12,13].

Utilization of healthcare services including immunization is a complex phenomenon involving health service providers, cultural beliefs, parent’s characteristics and socio-economic factors, and even the geographic barriers [14]. It is well recognized that full immunization coverage and broader socio-economic benefit of immunization are interrelated. However, such benefits of immunization are not yet achieved in many West African countries including Senegal, as the coverage of full immunization among children is below the national and international target and even not equally distributed across geographic regions and ethnic groups [8]. Therefore, it is crucial to identify the various factors that may influence children to be fully immunized. Although immunization coverage rate and associated factors are the emerging global health topics in many countries, limited studies are available in the context of Senegal. A number of studies have reported their findings on child immunization coverage but few of them generated evidence about the socio-demographic factors associated with full immunization among children. However, the existing studies focused on either specific target areas (e.g., urban, rural, geographic location), among younger aged children or on certain vaccine-preventable diseases [10,15,16]. Also existing studies are unable to present the current scenario of full immunization coverage as these studies often used either previous round data or not representative of the country scenario of full immunization status. Although a recent study served the need for documenting full immunization coverage and its determining factors to a great extend, however, this study also needs to monitor the change of immunization coverage and determining factors over time as it utilized the latest Demographic and Health Survey (DHS) 2017 dataset for analysis [8]. We hypothesize that the current immunization coverage rate and their determinants will potentially help to serve the purpose of launching Continuous-DHS program by providing up-to-date information to the policy makers for planning, monitoring, and evaluating national EPI program to manage vaccine-preventable diseases in an effective manner.

## 2. Materials and Methods

### 2.1. Data Source

For this study, dataset was extracted from the latest Continuous Senegal Demographic and Health Survey (EDS-Continue) 2017. The survey was a nationally representative cross-sectional household survey that provides up-to-date information on socio-demographic, maternal, and child health indications including individual-level vaccination coverage. Childhood immunization information was collected for surviving children aged up to three years. Immunization data were collected based on the records from vaccination cards, and maternal recall in those cases when mothers were unable to show vaccination record card. Face-to-face interviews with the reproductive-aged women (15–49 years) were conducted for collecting data using the structured questionnaire and based on the MEASURE DHS program model. A two-stage stratified random sampling technique was used for this survey. More about the sampling procedure and data collection technique has been described elsewhere [17]. All the DHS data are publicly accessible and were made available upon request by MEASURE DHS. Furthermore, approval was sought from and given by the MEASURE DHS program office to use this data set. According to the DHS, written informed consent was obtained from all participants before the interview. Children of 12 to 36 months of age were included in this analysis upon availability of immunization information in the dataset to capture the immunization status of the children. Children younger than 12 months were excluded from the analysis as they were not old enough to receive the full schedule of routine vaccines.

### 2.2. Outcome Variable

The outcome variable of the analysis was children’s immunization status and categorized as “fully immunized” and “partial/unimmunized.” Immunization status was categorized as “fully immunized” if the children had received the full doses of all the standard eight antigens; one dose of the vaccine against tuberculosis (BCG), three doses of pentavalent (DPT, Hib, and HepB), three doses of polio vaccine (OPV), and one dose of measles vaccines otherwise “partially/unimmunized” [18]. To obtain the outcome variable we first re-coded individual eight vaccines as complete and incomplete and then combined them by creating a dummy variable and coded as “1” for fully immunized and “0” otherwise. Like earlier study, vaccination status was considered as “not immunized” for those cases when vaccination record cards were unavailable and mothers indicated that they did not know about their children’s vaccination status since such responses reflect the negative response about immunization [19].

### 2.3. Explanatory Variables

Explanatory variables were selected based on the existing literature, prior knowledge, and availability of variables in the dataset and a framework was developed on the light of Anderson’s Behavior Health Model [14]. The model has been used widely to examine the factors associated with the utilization of health services including immunization uptake [19,20,21]. On the light of Anderson’s model and based on the literature, explanatory variables were grouped into three categories: external environment, predisposing factors, and enabling factors (Figure 1). The external environment includes area of residence, ecological zone, and ethnicity. Predisposing factors include sex of children, childbirth order, mothers’ age at child delivery, current marital status, parental (mother and father) education, and access to mass media. Enabling factors included wealth index, utilization of ANC and postnatal care (PNC), place of delivery, health care decision-maker, and distance of health facility. In this analysis, categorization of continuous variables was done in the light of previous literature. Self-reported parental educational attainment was used and categorized as “no education,” “primary,” “secondary,” and “higher.” No education refers to not attaining any formal education, primary is defined as completing grade 5, secondary as completing grade 10, and higher is defined as attaining more than grade 10. We had utilized the predetermined wealth index category provided in the dataset where it was generated from selected household assets using principal component analysis (PCA) and classified into five groups as: “poorest,” “poorer,” “middle,” “rich,” and “richest.” Decision-making ability of mothers’ health care was categorized into four groups as “herself,” “jointly with husband,” “husband alone,” and “other.” Further, childbirth order, family size, and access to mass media were also categorized and included as the explanatory variable in this analysis.

### 2.4. Statistical Analysis

In Senegal continuous DHS dataset vaccination information were not collected for the children more than 36 months of age, therefore they were not included in the current study. After dropping the missing information on immunization status a total of 4416 children remained and were included in the analysis. A proper sampling weight was used in this analysis to make the sample more representative of the population at the national level. Descriptive statistics such as the frequency with percentage were executed to represent the background characteristics of study participants, and proportion with 95% confidence interval (CI) was used for presenting the coverage rate of fully immunized children. Multivariable logistic regression models were constructed to observe the significant influencing factors for full-immunization with all the antigens and results were presented in terms of adjusted odds ratio (AOR) with a 95% confidence interval (CI). Before the execution of a multivariate regression model, bivariate analysis was also conducted to trace out the significant factors and then variables, significant at *p*-value ≤ 0.25 were included in the multivariate logistic regression model. Two separate logistic regression models; Model I and Model II were constructed to obtain the unadjusted odds ratio (from univariate analysis) and adjusted odds ratio (from multivariate analysis) and presented with 95% confidence interval (CI). Diagnostic tests were performed in the analysis. Variance inflation factor (VIF) was calculated and observed to detect multicollinearity problem in the regression model. Linear predicted value and linear predicted value squared were also checked to detect a specification error using *linktest* command in Stata. Moreover, the goodness of fit of the regression model was verified using Hosmer–Lemeshow test. All the data management procedures and statistical analysis were performed using Stata software version SE 14.0 (Stata Corporation, College Station, TX, USA).

## 3. Results

### 3.1. Characteristics of Study Participants

Table 2 provides the demographic and socioeconomic information of the study participants and the prevalence of full immunization coverage over background information. Among the study participants, 62.13% of the participants lived in rural areas, 32.82% were from western ecological zone of Senegal, and 37.02% belonged to the Wolof ethnic group. The proportion of male and female were almost equally distributed (50.73% and 49.27% respectively) while the majority of the children (80.37%) were born at healthcare facilities. Approximately 59% of mother received the recommended number of antenatal care (ANC) (four or more ANC) while 57.54% of mothers received standard postnatal care. Most of the households (60.50%) consisted of more than 10 members and 58.99% of the mothers and 71.17% of fathers had no formal education. Majority of the of the children’s mothers (70.48%) were aged between 20 to 34 years at the time of delivery. More than 81% of the mothers had access to mass media (radio and television), 23.22% of mothers were from the poorest quintile, and 18.06% were from the richest quintile. Only few percentage of mothers (5.71%) reported that they had the decision-making ability of their own healthcare where most of mothers (73.72%) depend on their husbands’ decision. Regarding the distance of facilities from their household, most of the mothers (71.91%) reported that distance of health facility was not a big problem for child immunization.

### 3.2. Vaccination Coverage Rate among Study Participants

Full immunization coverage rate of study participants over their background characteristics is also presented in Table 2. The study found that overall, 70.96% of the children were fully immunized. Full immunization coverage was higher in urban areas (76.51%) than the rural (67.57%) and the highest in west (80.00%) ecological zone and among Serer ethnic group (77.24%). Full immunization coverage was almost the same across sex of the children, slightly higher among second-born children (73.29%) and among the children who were born at healthcare facilities (74.03%). Findings also revealed that full immunization coverage was higher among the children of mothers who received at least four ANC and any PNC services (Table 2). Children of elderly mothers, smaller family size (less than 4 members), and marital status of the mother (e.g., living with partner) had the higher full immunization coverage. Full immunization coverage rate was also seen increasing along with the increasing of educational attainment of their parents and household wealth index. Full immunization coverage was found as 81.48% and 84.52% for the children whose mother and father had higher level of educational attainment respectively. In addition, findings revealed that full immunization coverage rate was 81.93% for the richest household while it was still 59.77% for the children from the poorest households. It was also observed that full immunization coverage was higher among the children of mothers who had the access of mass media, were the decider of their own health care, and reported that health facilities was not a problem compared to the counterparts. The descriptive statistics of vaccination coverage by type of vaccine over geographic regions are presented in Table 3 and full immunization coverage by geographic regions is reported in Figure 2.

From the Table 3, it can be seen that the coverage of BCG was the highest in the Kaolack region, whereas all the three doses of DPT (DPT 1, 2, and 3) and polio vaccine coverage were the highest in Thies region. Moreover, Dakar holds the highest coverage for measles vaccine. Figure 2 shows that full immunization coverage (all the recommended vaccines) was highest in Dakar (81.14%). On the contrary, Kedougou region had the lowest immunization coverage rate (29.22%) followed by Tambacounda region (44.77%).

### 3.3. Associated Factors for the Vaccination Coverage

Table 4 shows the potential factors that are associated with the full-immunization coverage among the children aged 12 to 36 months in terms of both the unadjusted (model I) and adjusted (model II) odds ratios. The analysis shows that several socio-demographic factors like ecological zone, ethnicity, place of delivery, utilization of ANC and PNC, maternal age, access to mass media, and wealth quintile were significantly associated with the full immunization status in both the unadjusted and adjusted models. Some other factors such as area of residence, childbirth order, parental education, health care decision-making, and distance of health facilities were significantly associated, as observed in the unadjusted model (model I). However, after adjusting all the possible variables in the adjusted model (model II) no such association was observed. 

From model I, it was observed that children were more likely to be fully immunized who were from urban areas (OR: 1.56; CI: 1.36–1.80; *p* < 0.001), west ecological zone (OR: 2.88; CI: 2.39–3.47; *p* < 0.001), and from Serer ethnic group (OR: 1.77; CI: 1.44–2.18; *p* < 0.001). Second-birth child and the children who were born at health facilities and belongs to the richest family had higher odds ratio for being fully immunized. We also observed that children of mothers who have access to the mass media, were the decider of self-health care, had higher educational attainment, and utilized ANC and PNC services had significantly higher odds than the counterparts (Table 4). Moreover, children of mothers having decision-making ability for their own health care and thought distance of health facility was not a big issues for child immunization were significantly found more likely to be fully immunized than the counterpart. 

Findings from the multivariate analysis (Table 4, model II) revealed that children who lived in the western zone of Senegal were 1.66 times (CI: 1.25–2.21; *p* = 0.001) more likely and the children of Serer ethnic group were 1.43 times (CI: 1.09–1.88; *p* = 0.011) more likely to be fully immunized than the children living in the southern zone and from the Poular ethnic group. Similarly, children whose mothers delivered in health facilities were more likely to be fully immunized than those whose mothers delivered at home (AOR = 1.47; CI: 1.20–1.80). In addition, mothers with four ANC visits during pregnancy were more likely to have their children fully immunized than those with no ANC visits (AOR: 2.06 CI: 1.19–3.57; *p* = 0.010) similarly, children of mothers who received PNC services were 1.25 times (CI: 1.05–1.49; *p* = 0.011) more likely to be fully immunized than their counterparts. Likewise, children of mothers aged 35 years or more were more likely to be fully immunized compared to those of <20 years of age, as well as the children of mothers who have the access to mass-media (AOR: 1.40 CI 1.15, 1.71; *p* = 0.010) compared to their counterparts.

## 4. Discussion 

Although substantial improvement have been made for protecting Senegalese children from violence, poverty, and child labor, children’s health still remain a serious public health problem [22]. Childhood immunization is one of the most effective public health interventions for diminishing the burden of disease among young children in resource-poor settings. The present study examined the status of full immunization coverage among 12–36 months aged children and determinants of full immunization status. The study observed that the overall full immunization coverage was 70.96% which is low compared to the global immunization coverage, but better than many African countries [23]. Several factors such as regional variation, ethnicity, number of ANC and PNC received, mother’s age at delivery, access to mass media, and wealth index have the significant association with full immunization status. There were also some other crucial factors such as, area of residence, birth order, place of delivery, parental education, and distance to health care facility which were significantly associated with the uptake of full dose vaccines independently. 

Despite the success, full immunization coverage varies across the country and among the type of vaccines. Our study observed a seven percentage point decline in coverage of immunization from BCG at birth (95.33%) to measles (88.69%) to be administrated after nine months of birth. Similar findings were also found in earlier studies, however, this coverage gap has been reduced over time [8,9]. This finding indicates the prevalence of missed opportunities and the challenge of introducing measles vaccine routinely after the ninth month of birth. Therefore, more promotional activities and actions are needed to increase the consciousness among the guardians for the uptake of all the essential vaccines. A recent study showed the consequence of low measles vaccine coverage in Europe despite of their socio-economic strata [24]. Reason behind lower uptake rate of measles vaccine may be because of the experience of adverse effect of earlier vaccines, failure to remember or understand the schedule of routine immunization, or because of less awareness or guidance from health professionals [25]. Our study found differences in full immunization coverage rate across different geographic regions and ethnic groups. For instance, children from the western region and children from the Serer ethnic group had the highest rate for full immunization coverage compared to others. Such geographical disparities in the coverage of full immunization were also found in many low and middle-income countries [19,26,27]. This difference may be due to various supply side factors such as distance, quantity and the quality of health-care services, fragile service delivery, and communication system, and also demand side factors such as, differences in the religious conservatism, fear of side effects, and the acceptance of immunization services based on cultural beliefs or differences [28]. Therefore, policy should target the low coverage regions considering the both supply and demand side barriers. 

Apart from the effect of geographic regions and ethnic groups, accessibility to maternal health-care services i.e., antenatal and postnatal care (ANC and PNC) was also found to be the influencing factors for full immunization coverage. Like earlier findings, we also observed that mothers who utilized recommended ANC and PNC services were most prone to fully vaccinate their children [8]. This may be explained by the fact that during the ANC and PNC visits, mothers receive enough positive information about childhood vaccination which make them confident about their child’s preventive health. Besides, from the unadjusted model, we observed that distance to health facility was also a significant factor for the uptake of all the recommended vaccines. This finding was confirmed by an earlier study from a developing country where it was shown that walking or travelling time, transport facility, and distance are the key factors that influence the utilization of healthcare services and walking distance more than 30 min diminishes the uptake rate of vaccines by one-third [20]. Full immunization rate was also found to increase with the increase of parental educational attainment and maternal age at the time of delivery. Like other studies in various settings, we also observed a positive association between the parental education and full immunization coverage, which indicate that knowledge and awareness on preventive care is crucial for the uptake of vaccines [19,29,30]. Therefore, community-based behavior change program such as broadcasting motivational and awareness-developing programs on radio, television, staging informative drama, and announcing through mikes should target the uneducated mothers for the better understanding of the beneficial role of immunization so that they will be encouraged to vaccinate their children timely. 

Wealth status of the households played a significant role in immunizing their children. Our results are in similar line with various studies indicating that the chance of being fully immunized is positively associated with households’ wealth status [19,29,30]. Although immunization services was completely free of charge, indirect cost of vaccination such as income loss and transportation cost might be incurred which appeared as an additional economic burden for the poorest households [31,32]. From the univariate analysis, we found that full immunization was better among urban children than the rural, which was confirmed by other earlier studies [19,33]. This is due to the large number of quality health facilities that are available in most of the urban areas compared to rural, however, such relationship was not observed in multivariate analysis.

In accordance with various studies in different settings, healthcare decision-making ability was found as an important factor for full immunization coverage [19]. We have found that, if mothers are the decider of their self-healthcare then the children are more likely to be fully immunized and vice versa. This might be due to the health consciousness and autonomy in decision-making that triggers mothers to be concerned about their children’s health care and so as vaccine uptake also. Thus, community-based behavior change programs targeting parents might be helpful to be aware of their child immunization.

The study has several limitations. The study is based on secondary data and for few cases immunization records were obtained as the mother stated. Therefore, the potential effect of recall bias on our results cannot be ignored. Nonetheless, mother’s report is considered as the valid measure of immunization coverage in the absence of a health card [34]. Further, the explanatory variables were selected based on similar previous studies and upon availability in DHS dataset, however, there might be some other potential predictors of full immunization which was unable to capture. Despite such limitations, current study is based on the latest continuous nationally representative survey which presented the national representative immunization scenario of Senegalese children. Therefore, the findings are still noteworthy and relevant in drawing attention to the health policy makers for ensuring the benefit of vaccination for the betterment of child health.

## 5. Conclusions

Our study found that overall, 70.96% of the children were fully immunized which varied across regions, ethnic groups, and demographic characteristics of Senegalese children. Findings revealed that ecological zone, ethnicity, place of delivery, ANC and PNC service utilization, maternal age, mass media access, and wealth index were significantly associated with full immunization, which should be taken into account for mapping effective immunization program. Regular monitoring and evaluation of immunization coverage at low perfuming regions is necessary so that the broader benefit of immunization program can be achieved by all strata of the society. 

## Figures and Tables

**Figure 1 medicina-55-00480-f001:**
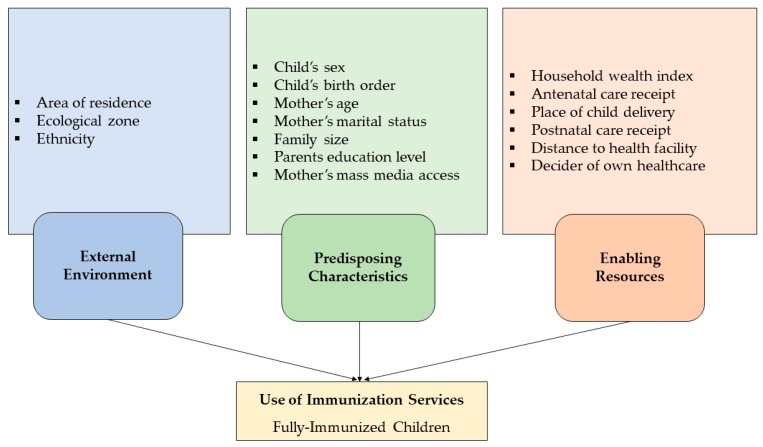
Theoretical framework of factors potentially associated with immunization coverage of children in Senegal (adopted from Anderson’s Behavior Health Model [14]).

**Figure 2 medicina-55-00480-f002:**
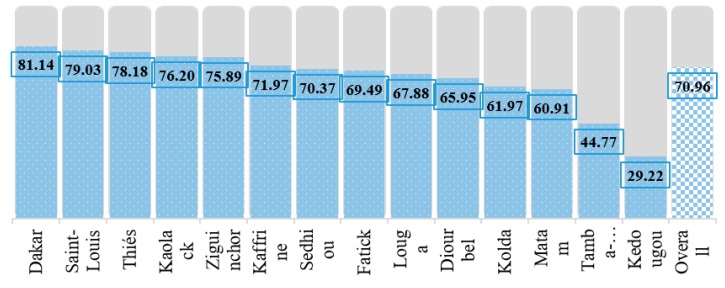
Full immunization coverage (%) by geographic regions of Senegal.

**Table 1 medicina-55-00480-t001:** Recommended age for BCG, VPO, Penta, Rota, Pneumo, VPO, MR, and AA vaccinations for children in Senegal and as per the WHO guidelines for childhood vaccinations.

Vaccine	Recommended Schedule with the National EPI Program	WHO Recommended Time Range
BCG + VPO zero	At birth	Birth–4weeks
Penta 1 + Pneumo 1 + VPO1 + Rota1	6 weeks	4 weeks–2 months
Penta 2 + Pneumo 2 + VPO2+ Rota2	10 weeks	8 weeks–4 months
Penta 3 + Pneumo 3 + VPO3	14 weeks	12 weeks–6 months
MR1 + VAA	9 months	38 weeks–12 months
MR2	15 months	15–19 months

Penta: pentavalent vaccine (DTC HiB, Hepatitis B); pneumo: pneumococcal conjugate vaccine; VAA: vaccine antimalarial (yellow fever); RR: measle rubella; BCG: bacillus of Calmette and Guerin; VPO zero: polio vaccine at birth; VPO1: oral polio vaccine.

**Table 2 medicina-55-00480-t002:** Sample distribution and full immunization coverage among children aged 12–36 months.

Characteristics of Sample	Frequency (%)	Full Immunization Coverage [95% CI]
**Immunization status**		
*Partially immunized*	1283 (29.04)	-
*Fully immunized*	3133 (70.96)	-
**External Environment**		
**Area of residence**		
*Urban*	1672 (37.87)	76.51 [74.42, 78.48]
*Rural*	2744 (62.13)	67.57 [65.79, 69.29]
**Ecological zone**		
*North*	738 (16.70)	70.30 [66.90, 73.49]
*West*	1449 (32.82)	80.00 [77.85, 81.98]
*Central*	1353 (30.63)	69.94 [67.44, 72.33]
*South*	877 (19.85)	58.13 [54.83, 61.36]
**Ethnicity**		
*Poular*	1238 (28.04)	65.74 [63.05, 68.34]
*Wolof*	1635 (37.02)	74.95 [72.79, 76.99]
*Serer*	747 (16.92)	77.24 [74.09, 80.11]
*Other*	796 (18.02)	64.96 [61.58, 68.20]
**Predisposing Characteristics**		
**Sex**		
*Male*	2240 (50.73)	71.79 [69.89, 73.62]
*Female*	2176 (49.27)	70.09 [68.13, 71.98]
**Birth order**		
*First*	1043 (23.61)	71.61 [68.79, 74.27]
*Second*	1522 (34.47)	73.29 [71.01, 75.45]
*Third or more*	1851 (41.92)	68.67 [66.52, 70.74]
**Mother’s age at delivery**		
*<20 years*	546 (12.36)	63.81 [59.69, 67.74]
*20–34 years*	3113 (70.48)	71.86 [70.25, 73.41]
*35 years and more*	758 (17.16)	72.38 [69.09, 75.46]
**Current marital status**		
*Married or living with partner*	4142 (93.79)	71.27 [69.87, 72.63]
*Other (widowed/divorced etc.)*	274 (6.21)	66.16 [60.35, 71.53]
**Family size**		
*less than 4*	71 (1.61)	74.66 [63.25, 83.46]
*4–6*	573 (12.98)	73.67 [69.90, 77.11]
*7–10*	1100 (24.91)	68.92 [66.12, 71.59]
*More than 10*	2672 (60.50)	71.11 [69.36, 72.80]
**Mother’s education ^a^**		
*No formal education*	2605 (59)	68.14 [66.32, 69.90]
*Primary*	1027 (23.26)	73.43 [70.64, 76.05]
*Secondary*	656 (14.87)	76.26 [72.85, 79.36]
*Higher*	127 (2.88)	81.48 [73.73, 87.33]
**Father’s education ^a^**		
*No formal education*	3143 (71.17)	68.14 [66.49, 69.75]
*Primary*	611 (13.84)	76.06 [72.51, 79.28]
*Secondary*	448 (10.14)	77.27 [73.15, 80.92]
*Higher*	214 (4.84)	84.52 [79.01, 88.78]
**Mass media access**		
*Yes*	3583 (81.13)	73.11 [71.63, 74.54]
*No*	833 (18.87)	61.69 [58.33, 64.93]
**Enabling Resources**		
**Place of delivery**		
*Home*	867 (19.63)	58.35 [55.03, 61.59]
*Institution*	3549 (80.37)	74.03 [72.57, 75.45]
**No. of ANC received ^b^**		
*No ANC*	70 (1.85)	38.23 [27.61, 50.12]
*1–3*	1487 (39.13)	71.28 [68.92, 73.52]
*4 and more*	2243 (59.02)	74.01 [72.15, 75.78]
**Received any PNC ^c^**		
*No*	1875 (42.46)	64.25 [62.05, 66.39]
*Yes*	2541 (57.54)	75.9 [74.20, 77.53]
**Wealth index**		
*Poorest*	1025 (23.22)	59.77 [56.74, 62.74]
*Poorer*	968 (21.92)	70.22 [67.26, 73.02]
*Middle*	910 (20.61)	71.46 [68.44, 74.31]
*Richer*	715 (16.19)	75.1 [71.79, 78.13]
*Richest*	797 (18.06)	81.93 [79.1, 84.45]
**Mother’s healthcare decision-maker**		
*Herself*	237 (5.71)	78.68 [72.98, 83.45]
*Jointly with husband*	771 (18.61)	70.93 [67.62, 74.03]
*Husband alone*	3054 (73.72)	70.92 [69.28, 72.50]
*By other*	81 (1.95)	66.24 [55.23, 75.73]
**Distance of health facility is a problem**		
*Yes*	1240 (28.09)	63.35 [60.62, 65.98]
*No*	3176 (71.91)	73.93 [72.37, 75.43]

CI: confidence interval; ^a^ no education, primary, secondary, and higher education refers to not attaining any formal education, completing grade 5, grade 10, and completing higher than grade 10, respectively; ^b^ ANC: antenatal care; ^c^ PNC: postnatal care.

**Table 3 medicina-55-00480-t003:** Vaccination coverage rate by each individual vaccines across geographic regions in Senegal.

Regions	BCG	DPT 1	DPT 2	DPT 3	Polio 1	Polio 2	Polio 3	Measles
	Coverage (95% CI)	Coverage (95% CI)	Coverage (95% CI)	Coverage (95% CI)	Coverage (95% CI)	Coverage (95% CI)	Coverage (95% CI)	Coverage (95% CI)
Dakar	99.52 (98.77, 99.82)	99.52 (98.77, 99.82)	99.18 (98.32, 99.61)	98.33 (97.24, 98.99)	96.49 (95.06, 97.52)	96.49 (95.06, 97.52)	83.95 (81.39, 86.22)	96.30 (94.84, 97.36)
Diourbel	95.44 (93.40, 96.87)	94.06 (91.81, 95.72)	92.93 (90.52, 94.75)	89.93 (87.20, 92.14)	95.70 (93.70, 97.08)	92.72 (90.29, 94.58)	73.29 (69.52, 76.74)	87.00 (83.99, 89.50)
Fatick	98.72 (96.01, 99.6)	98.82 (96.14, 99.65)	98.51 (95.73, 99.49)	95.83 (92.27, 97.79)	97.96 (95.01, 99.18)	97.46 (94.36, 98.88)	74.92 (68.83, 80.15)	91.44 (87.00, 94.46)
Kaffrine	96.15 (92.87, 97.96)	97.57 (94.68, 98.91)	95.52 (92.09, 97.51)	94.29 (90.59, 96.59)	96.38 (93.15, 98.11)	94.98 (91.43, 97.11)	79.51 (73.99, 84.11)	89.44 (84.93, 92.72)
Kaolack	99.71 (97.70, 99.96)	99.50 (97.58, 99.90)	97.81 (95.39, 98.97)	95.23 (92.17, 97.13)	98.86 (96.78, 99.60)	96.69 (93.97, 98.21)	82.53 (77.84, 86.41)	90.98 (87.19, 93.73)
Kedougou	61.90 (48.70, 73.55)	68.10 (54.94, 78.89)	64.89 (51.69, 76.15)	59.77 (46.60, 71.66)	68.06 (54.91, 78.85)	63.70 (50.49, 75.12)	42.94 (30.75, 56.04)	49.24 (36.53, 62.04)
Kolda	91.88 (87.63, 94.76)	95.44 (91.89, 97.48)	93.21 (89.19, 95.80)	91.91 (87.67, 94.78)	93.96 (90.09, 96.38)	92.19 (87.99, 95.00)	68.93 (62.72, 74.52)	87.21 (82.30, 90.91)
Louga	94.55 (91.22, 96.66)	95.91 (92.88, 97.68)	95.07 (91.86, 97.06)	91.84 (88.05, 94.51)	93.60 (90.10, 95.92)	91.31 (87.43, 94.07)	77.96 (72.78, 82.40)	84.1 (79.38, 87.90)
Matam	91.90 (86.88, 95.1)	94.10 (89.54, 96.75)	92.25 (87.30, 95.37)	89.55 (84.13, 93.26)	86.68 (80.86, 90.92)	84.39 (78.30, 89.01)	70.40 (63.29, 76.63)	81.83 (75.47, 86.82)
Saint-Louis	99.20 (97.02, 99.79)	99.18 (97.00, 99.78)	98.34 (95.89, 99.34)	97.74 (95.10, 98.97)	95.97 (92.86, 97.76)	93.92 (90.38, 96.22)	82.91 (77.96, 86.94)	94.95 (91.61, 97.01)
Sedhiou	95.77 (91.21, 98.01)	97.16 (93.01, 98.87)	94.89 (90.10, 97.43)	89.74 (83.91, 93.62)	93.51 (88.41, 96.46)	90.25 (84.50, 94.01)	77.40 (70.19, 83.29)	88.96 (83.01, 93.00)
Tamba-counda	72.96 (67.45, 77.84)	78.17 (72.94, 82.63)	72.39 (66.85, 77.31)	70.51 (64.90, 75.56)	76.50 (71.17, 81.10)	71.28 (65.70, 76.28)	56.97 (51.10, 62.65)	65.5 (59.73, 70.84)
Thiés	98.89 (97.59, 99.49)	99.70 (98.65, 99.93)	99.70 (98.65, 99.93)	98.48 (97.05, 99.22)	98.60 (97.21, 99.30)	98.6 (97.21, 99.30)	85.05 (81.85, 87.77)	92.52 (90.03, 94.43)
Ziguinchor	99.69 (94.50, 99.98)	98.67 (94.69, 99.67)	97.37 (93.04, 99.04)	94.57 (89.45, 97.28)	96.05 (91.32, 98.25)	93.71 (88.38, 96.68)	79.94 (72.61, 85.69)	92.00 (86.31, 95.45)
Overall	95.33 (94.66, 95.91)	96.11 (95.50, 96.64)	94.82 (94.12, 95.43)	92.83 (92.03, 93.55)	94.37 (93.65, 95.02)	92.69 (91.88, 93.42)	77.47 (76.22, 78.68)	88.69 (87.72, 89.59)

**Table 4 medicina-55-00480-t004:** Unadjusted and adjusted effects of factors that are associated with full immunization coverage.

Characteristics	Model I		Model II	
	Unadjusted OR (95% CI)	*p*-Value	Adjusted OR (95% CI)	*p*-Value
**External Environment**				
**Area of residence**				
*Urban*	1.56 (1.36, 1.80)	**<0.001**	0.96 (0.77, 1.21)	0.752
*Rural*	1.00		1.00	
**Ecological zone**				
*North*	1.71 (1.39, 2.10)	**<0.001**	1.42 (1.09, 1.84)	**0.009**
*West*	2.88 (2.39, 3.47)	**<0.001**	1.66 (1.25, 2.21)	**0.001**
*Central*	1.68 (1.4, 2.00)	**<0.001**	1.14 (0.89, 1.46)	**0.288**
*South*	1.00		1.00	
**Ethnicity**				
*Poular*	1.00		1.00	
*Wolof*	1.56 (1.33, 1.83)	**<0.001**	1.15 (0.93, 1.42)	0.197
*Serer*	1.77 (1.44, 2.18)	**<0.001**	1.43 (1.09, 1.88)	**0.011**
*Other*	0.97 (0.80, 1.16)	0.719	0.86 (0.68, 1.10)	0.232
**Predisposing Characteristics**				
**Sex**				
*Male*	1.09 (0.95, 1.24)	0.212	1.08 (0.93, 1.26)	0.314
*Female*	1.00		1.00	
**Birth order**				
*First*	1.15 (0.97, 1.36)	0.098	1.25 (0.95, 1.64)	0.108
*Second*	1.25 (1.08, 1.45)	**0.003**	1.17 (0.97, 1.43)	0.103
*Third or more*	1.00		1.00	
**Mother’s age at delivery**				
*<20 years*	1.00		1.00	
*20–34 years*	1.45 (1.20, 1.75)	**<0.001**	1.45 (1.10, 1.93)	**0.009**
*35 years and more*	1.49 (1.17, 1.88)	**0.001**	1.57 (1.10, 2.23)	**0.012**
**Mother’s education**				
*No formal education*	1.00		1.00	
*Primary*	1.29 (1.10, 1.52)	**0.002**	0.91 (0.74, 1.12)	0.361
*Secondary*	1.5 (1.23, 1.83)	**<0.001**	1.21 (0.91, 1.61)	0.199
*Higher*	2.06 (1.31, 3.24)	**0.002**	0.68 (0.39, 1.20)	0.184
**Father’s education**				
*No formal education*	1.00		1.00	
*Primary*	1.49 (1.22, 1.81)	**<0.001**	1.05 (0.82, 1.34)	0.680
*Secondary*	1.59 (1.26, 2.01)	**<0.001**	1.06 (0.80, 1.42)	0.686
*Higher*	2.55 (1.75, 3.73)	**<0.001**	1.48 (0.93, 2.35)	0.102
**Mass media access**				
*Yes*	1.69 (1.44, 1.98)	**<0.001**	1.40 (1.15, 1.71)	**0.001**
*No*	1.00		1.00	
**Enabling Resources**				
**Wealth index**				
*Poorest*	1.00		1.00	
*Poorer*	1.59 (1.32, 1.91)	**<0.001**	1.26 (1.00, 1.59)	**0.049**
*Middle*	1.69 (1.39, 2.04)	**<0.001**	1.05 (0.81, 1.38)	0.704
*Richer*	2.03 (1.64, 2.50)	**<0.001**	1.09 (0.78, 1.53)	0.597
*Richest*	3.05 (2.45, 3.80)	**<0.001**	1.34 (0.91, 1.99)	0.140
**No. of ANC received**				
*No ANC*	1.00		1.00	
*1–3*	4.01 (2.45, 6.57)	**<0.001**	2.27 (1.32, 3.90)	**0.003**
*4 and more*	4.60 (2.82, 7.51)	**<0.001**	2.06 (1.19, 3.57)	**0.010**
**Place of delivery**				
*Home*	1.00		1.00	
*Institution*	2.04 (1.74, 2.38)	<**0.001**	1.47 (1.20, 1.80)	**<0.001**
**Received any PNC**				
*No*	1.00		1.00	
*Yes*	1.75 (1.54, 2.00)	**<0.001**	1.25 (1.05, 1.49)	**0.011**
**Mother’s healthcare decision-maker**				
*Husband alone*	1.00		1.00	
*Herself*	1.51 (1.10, 2.09)	**0.011**	1.30 (0.90, 1.87)	0.155
*Jointly with husband*	1.00 (0.84, 1.19)	0.993	0.89 (0.73, 1.09)	0.271
*By other*	0.80 (0.5, 1.28)	0.362	0.81 (0.49, 1.36)	0.433
**Distance of health facility is a problem**				
*Yes*	1.00		1.00	
*No*	1.64 (1.43, 1.89)	**<0.001**	1.13 (0.94, 1.37)	0.194

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
