# Peer review of "Coverage and Determinants of Full Immunization: Vaccination Coverage among Senegalese Children"

_medicina, 2019, doi:10.3390/medicina55080480_

Round 1

Reviewer 1 Report

the fact that the study is based on nation-wide data makes it very powerful and suitable for international comparisons

the  wide range of explanatory variables used in the study add to the quality of the data. However, I am not very much fond of picking on ethnic groups as a variable unless it plays a major role irrespective of geographical/ecological regions.

the quality of the language is not consistent, needs some professional assistence. For example you mention often "fully immunization" instead of " fully immunized" or "full immunation"....

What does the third column in table 1 -"prevalence (95% CI)" - represent? Please describe in the title or as a footnote or correct

Author Response

Comment 1: The fact that the study is based on nation-wide data makes it very powerful and suitable for international comparisons the wide range of explanatory variables used in the study add to the quality of the data. However, I am not very much fond of picking on ethnic groups as a variable unless it plays a major role irrespective of geographical/ecological regions. 

Response: Authors like to thank the reviewer for their kind review and valuable comments. Ethnic group was selected in this analysis as an explanatory variable as it was mentioned in earlier literature that, acceptance of vaccines varies across different ethnic groups and regions. Life style, social and religious norms and other affects such variations. Our intention was to identify the variations of full vaccination rate among different groups including geographical and ethnic groups. In regression table (table 3), we also found significant variation of full immunization coverage among both the geographical/ecological regions and ethnic groups.

Comment 2: The quality of the language is not consistent, needs some professional assistance. For example you mention often "fully immunization" instead of “fully immunized" or "full immunization"....

Response: Thank to the reviewer for the concern. The overall English language of the manuscript has been checked and corrected now by native English speaker as suggested. Please see the revised version.

Comment 3: What does the third column in table 1 -"prevalence (95% CI)" - represent? Please describe in the title or as a footnote or correct

Response: The term “prevalence” is now altered to “full vaccination coverage”. Please see table 2 of the revised version.

Reviewer 2 Report

This paper aims to present factors associated with vaccine coverage in a classic analysis of DHS survey data. the analysis (and the findings) are very similar to previosu studies both conducted in neighbouring countrie sand in Senegal itself. In particular it is very similar to a 2017 paper by Mouhamed Abdou Salam Mbengue et al  (ref 8 in the paper). Albeit using more recent data. The paper would need to be more explicit about what it adds to the literature, what is the added value since the 2017 paper and an analysis of how the situation in Senegal has changed since 2010-11 which was thebasis of the 2017 paper. there are also concerns about perhaps over interpreting the data at a level for which the data collection was not powere for. More importantly there is a concern about no involvement from any Senegalese authors. While I understand this is a secondary analysis of previsouly collected data this raises ethical issues about including individuals involved in the primary data collection and would also help the authors gain further insight into the data and its limitations. The level of English is also not sufficient for publication at this stage

Despite those major limitation the paper has potential value and therefore if these points can be addressed through a major revision, the paper should be reconsidered.

Minor points:

1) Could a table with the schedule in Senegal be included for clarity?

2) The authors claim that" The existing studies focused on either specific target areas  and age group or on a certain vaccine-preventable diseases while the studies often focused on only individual characteristics. Also existing research unable to represent the current scenario of full immunization coverage as these studies often used previous data and not representative of the country scenario of full immunization status."

 These claims are not entirely accurate. The proposed study examines almost exactly the same variables as Mouhamed Abdou Salam Mbengue et al 2017. (ref 8). The claim that the previous studies use retrospective also applies to this study and is inherent to using DHS data which is by definition going to be retrospective at the point the data is analysed. The authors need to make a strong claim as to what their study adds to the exisiting literature- besides using more up to date data.

Line 40: "Vaccines have the biggest success stories in the field of modern medicine"

What does this mean? Peraps in the field of public health is more appropriate

Line 54: not sure what “adaptation of EPI” means can the authors clarify?

 Line 73 a rise in measles outbreak do the authors mean a rise in measles incidence?

 Line 79: Senegal is in West Africa, not South Africa

Line 222L regional coverage is offered without any confidence interval. Was the DHS survey powere to determine coverage at this level of granularity? If no this should be made clear and precision in the form of confidence intervals should be offered this is not mentioned in the limitations?

Author Response

Comment: This paper aims to present factors associated with vaccine coverage in a classic analysis of DHS survey data. The analysis (and the findings) are very similar to previous studies both conducted in neighboring countries sand in Senegal itself. In particular it is very similar to a 2017 paper by Mouhamed Abdou Salam Mbengue et al (ref 8 in the paper). Albeit using more recent data. The paper would need to be more explicit about what it adds to the literature, what is the added value since the 2017 paper and an analysis of how the situation in Senegal has changed since 2010-11 which was the basis of the 2017 paper. There are also concerns about perhaps over interpreting the data at a level for which the data collection was not powered for. More importantly there is a concern about no involvement from any Senegalese authors. While I understand this is a secondary analysis of previously collected data this raises ethical issues about including individuals involved in the primary data collection and would also help the authors gain further insight into the data and its limitations. The level of English is also not sufficient for publication at this stage

Despite those major limitation the paper has potential value and therefore if these points can be addressed through a major revision, the paper should be reconsidered.

Response: Authors like to thank the reviewer for the valuable time to review and make valuable comments on this manuscript. Authors agree with the reviewer that the current paper is seems to be close to an earlier published work in 2017 by Mouhamed Abdou Salam Mbengue et al  using 2010–2011 DHS data of Senegal. The current study utilized the latest DHS data, therefore this findings may indicate whether there is any improvement in immunization status among children in Senegal over the period. Further this study included broader age group (12 to 36 months of age) and also included more explanatory variables in light of Anderson’s Behavior Health Model (reff-14) and reported the coverage of individual vaccines by 14 administrative regions. The way how this current study add values are now inserted in the introduction section. Please see lines 94-104 in track change version. While this dataset is publicly available and the authors sought permission and approved from MEASURE DHS to utilize this data, therefore authors did not include any personnel related to primary data collection, albeit it is expected that such involvement might help to gain more insights to interpret the findings. The overall English language of the manuscript has been checked and corrected now by native English speaker as suggested. Please see the revised version of the manuscript.

Minor points

Comment 1) Could a table with the schedule in Senegal be included for clarity?

Response: Authors thanks the reviewer for the valuable comment. Agreeing with the reviewer concern, a table describing immunization schedule in Senegal are now inserted. Please see table 1 of the revised version of the manuscript.

Comment 2) The authors claim that "The existing studies focused on either specific target areas and age group or on a certain vaccine-preventable diseases while the studies often focused on only individual characteristics. Also existing research unable to represent the current scenario of full immunization coverage as these studies often used previous data and not representative of the country scenario of full immunization status."

These claims are not entirely accurate. The proposed study examines almost exactly the same variables as Mouhamed Abdou Salam Mbengue et al 2017. (Ref 8). The claim that the previous studies use retrospective also applies to this study and is inherent to using DHS data which is by definition going to be retrospective at the point the data is analyzed. The authors need to make a strong claim as to what their study adds to the existing literature- besides using more up to date data.

Response: Authors agree with the reviewer concern. The statements that are in reviewer concern are now rephrased to remove the inconsistencies. Further the importance of this paper is now justified in the introduction section. Please see line 83 to 103 in page 2 and 3 in the revised version.

Comment 3) Line 40: "Vaccines have the biggest success stories in the field of modern medicine" What does this mean? Perhaps in the field of public health is more appropriate

 Response: Authors agree with the reviewer concern. The sentence has now rephrased as suggested. Please see page 1 line 41

Comment 4) Line 54: not sure what “adaptation of EPI” means can the authors clarify?

Response: The term adaptation has now been altered to implementation to make it clearer. Please see line 54

Comment 5) Line 73 a rise in measles outbreak do the authors mean a rise in measles incidence?

Response: Here, “a rise in measles outbreak” mentions the rise in measles incidence. The term “outbreak” is now replaced to “incidence” for make clearer. Please see line 73

Comment 6) Line 79: Senegal is in West Africa, not South Africa

Response: Author apologize for thin unexpected mistake. This has now corrected. Please see line 79

Comment 7) Line 222L regional coverage is offered without any confidence interval. Was the DHS survey powered to determine coverage at this level of granularity? If no this should be made clear and precision in the form of confidence intervals should be offered this is not mentioned in the limitations?

Response: Authors agree with reviewer concern. Confidence intervals are now mentioned to express the coverage rate of vaccines. Please see the updated table 3 in the revised version.